# Examining geospatial and temporal distribution of invasive non-typhoidal *Salmonella* disease occurrence in sub-Saharan Africa: a systematic review and modelling study

Jong-Hoon Kim ,[1] Bieke Tack ,[2,3] Fabio Fiorino,[4,5] Elena Pettini,[4] Christian S Marchello,[6] Jan Jacobs,[3,7] John A Crump ,[8] Florian Marks ,[9,10] Vacc-iNTS Consortium

For numbered affiliations see end of article.

**Correspondence to**
Dr Jong-Hoon Kim;
jonghoon.kim@ivi.int

## ABSTRACT

**Background** Invasive non-typhoidal *Salmonella* (iNTS) disease is a significant health concern in sub-Saharan Africa. While our knowledge of a larger-scale variation is growing, understanding of the subnational variation in iNTS disease occurrence is lacking, yet crucial for targeted intervention.

**Method** We performed a systematic review of reported occurrences of iNTS disease in sub-Saharan Africa, consulting literature from PubMed, Embase and Web of Science published since 2000. Eligibility for inclusion was not limited by study type but required that studies reported original data on human iNTS diseases based on the culture of a normally sterile site, specifying subnational locations and the year, and were available as full-text articles. We excluded studies that diagnosed iNTS disease based on clinical indications, cultures from non-sterile sites or serological testing. We estimated the probability of occurrence of iNTS disease for sub-Saharan Africa on 20 km × 20 km grids by exploring the association with geospatial covariates such as malaria, HIV, childhood growth failure, access to improved water, and sanitation using a boosted regression tree.

**Results** We identified 130 unique references reporting human iNTS disease in 21 countries published from 2000 through 2020. The estimated probability of iNTS occurrence grids showed significant spatial heterogeneity at all levels (20 km × 20 km grids, subnational, country and subregional levels) and temporal heterogeneity by year. For 2020, the probability of occurrence was higher in Middle Africa (0.34, 95% CI: 0.25 to 0.46), followed by Western Africa (0.33, 95% CI: 0.23 to 0.44), Eastern Africa (0.24, 95% CI: 0.17 to 0.33) and Southern Africa (0.08, 95% CI: 0.03 to 0.11). Temporal heterogeneity indicated that the probability of occurrence increased between 2000 and 2020 in countries such as the Republic of the Congo (0.05 to 0.59) and Democratic Republic of the Congo (0.10 to 0.48) whereas it decreased in countries such as Uganda (0.65 to 0.23) or Zimbabwe (0.61 to 0.37).

**Conclusion** The iNTS disease occurrence varied greatly across sub-Saharan Africa, with certain regions being disproportionately affected. Exploring regions at high risk

## STRENGTHS AND LIMITATIONS OF THIS STUDY

⇒ We examined the geospatial and temporal distribution of the probability of occurrence of invasive non-typhoidal *Salmonella* (iNTS) disease for sub-Saharan Africa on 20 km × 20 km grids.
⇒ Occurrence data points are few with cumulative occurrences from 2000 through 2020 only representing around 3% of the 20 km × 20 km grids of sub-Saharan Africa.
⇒ iNTS disease represents only a fraction of the transmission of the pathogen and thus it is likely that the estimates of the occurrence of iNTS disease underestimate the transmission of the pathogen.

for iNTS disease, despite the limitations in our data, may inform focused resource allocation. This targeted approach may enhance efforts to combat iNTS disease in more affected areas.

## INTRODUCTION

Non-typhoidal *Salmonella* (NTS) is a leading cause of invasive bacterial disease in sub-Saharan Africa (sSA). Invasive NTS (iNTS) disease was estimated to have caused 535 000 cases and 77 500 deaths in 2017.[1] More than three-quarters of these cases and deaths occurred in sSA, where the incidence rates were estimated at 44.8 per 100 000 person-years and case fatality was estimated at 17.1%.[2 3]

The geographical predominance of iNTS disease in sSA can be partially explained by the prevalence of key host risk factors and the transmission of human-adapted NTS in the region.[4 5] While NTS strains circulating in other settings typically cause self-limiting foodborne diarrhoeal disease, genetic adaptations observed in the African strains

resulted in increased invasive disease[6] and possibly the establishment of a human reservoir.[7] Impaired immunity due to young age, recent or current *Plasmodium falciparum* malaria infection, anaemia, malnutrition and HIV infection renders the sSA population susceptible to iNTS disease, in particular among infants and children under the age of 5 years.[4] In addition to these host risk factors, environmental risk factors for iNTS disease such as unimproved water sources contribute to the high incidence of iNTS disease in sSA.[7]

While existing studies provide estimates at the country, regional and global levels mainly based on the incidence rates from population-based surveillance studies,[1 2 8] they fail to provide geospatial heterogeneity of iNTS disease at the subnational level, which is crucial for prioritising areas for targeted iNTS surveillance and control programmes. We aimed to characterise the geospatial variation in the probability of iNTS occurrence across sSA at a resolution of 20 km × 20 km using iNTS disease occurrences reported in the literature and geospatial data of host and environmental iNTS risk factors.

## METHODS

### iNTS occurrence data

We sought data on iNTS occurrence and incidence rates in the peer-reviewed literature available in PubMed, Embase and Web of Science published from 1 January 2000 through 30 June 2021 in the English language. We designed the search strings to account for each database and varying usage of the terms, non-typhoidal *Salmonella*, and to restrict the topic to human infections in sSA while adapting to the syntax of different databases (online supplemental material S1). We conducted the review according to the Preferred Reporting Items for Systematic review and Meta-Analysis (PRISMA).[9] We used Rayyan[10] for a collaborative review of the literature and collated the data in Google Sheets. The PRISMA flowchart shows an overview of the literature review process (figure 1).

Three authors (BT, FF and EP) reviewed the literature and collated the data with at least two independent reviewers reviewing each article. Disagreements between the reviewers were resolved after discussion with the reviewers and the first author (J-HK). Studies were

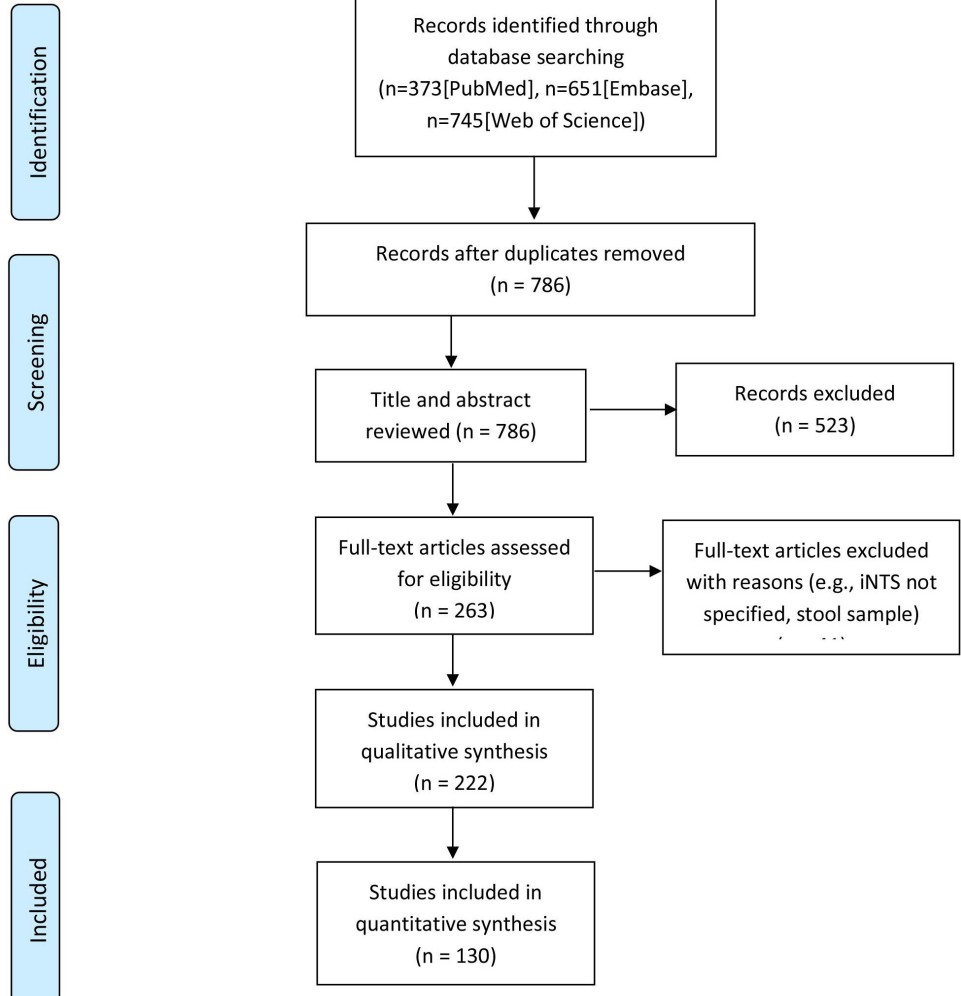

**Figure 1** Preferred Reporting Items for Systematic review and Meta-Analysis flow diagram for the systematic review of geospatial distribution of invasive non-typhoidal Salmonella (iNTS) occurrence in sub-Saharan Africa, 2000–2020.

eligible regardless of the study type (eg, outbreak investigation, population-based surveillance or case report) if they reported original findings of human iNTS disease defined as isolation of NTS from a normally sterile site (eg, blood, cerebrospinal fluid, bone marrow) in a patient with subnational location and year as a full-text article. Conference proceedings were excluded. We excluded studies that used clinical indication (ie, symptoms and signs), culture of a non-sterile site (eg, stool) or serology to classify an iNTS disease. We excluded studies that did not report iNTS separately from *Salmonella typhi* or *Salmonella paratyphi* A, B or C. Abstracted variables include year, location, diagnostic method and the number of iNTS disease cases. If observations spanned multiple years, the year of occurrence for each iNTS disease case was selected uniformly at random among observation years, which leads to the same number of cases for each year on average.

### Geo-positioning of iNTS occurrence

iNTS occurrence was defined as a 20 km × 20 km grid cell for which the spatial resolution was chosen by taking into account the feasibility of exploring subnational heterogeneity, computational cost and a previous study of a similar resolution.[11] The position was defined as a grid cell to which the longitude and latitude of the area belong based on Google Maps (Alphabet, Mountain View, California, USA). If the record reported the subnational area in which iNTS was reported was smaller than a grid cell, we assumed that the iNTS disease occurred in the grid cell. On the other hand, if the subnational administrative units included more than one grid cell, we assumed that iNTS disease could occur on any of the grid cells within the boundary and used the covariate value averaged over the grid cells that fall on the area. If the occurrence was reported in a hospital, the grid cell for occurrence was selected randomly assuming that the probability that the iNTS disease appeared at $d$ km from the hospital is $de^{-kd}$, where the rate of exponential decay $k$ was chosen differentially according to the catchment area of the hospital. The exponential decay of the probability was based on the idea that the frequency of visits to a hospital decreases with the distance, more generally known as distance decay, from the hospital following an exponential decay.[12] We assumed that the catchment area was 20 km and 100 km for primary and secondary or tertiary hospitals, respectively (ie, $k$ was 0.05 and 0.01 for primary and secondary/tertiary hospitals, respectively), which implies that 63% of the cases occurred within the boundary of 20 km and 100 km, respectively, and around 86% of the cases occurred within the boundary of 40 km and 200 km, respectively. We conducted a sensitivity analysis by assuming an average distance of 50 km and 200 km for the iNTS occurrence location from the tertiary hospital (ie, 0.02 or 0.005 for $k$), respectively.

### Geospatial covariates

Our study investigated factors that may affect the risk of iNTS disease across sSA, focusing on a variety of hosts[13] and environmental factors.[7] We examined these factors at a detailed resolution of at least 20 km × 20 km across the continent. For host factors, we looked into the prevalence of HIV (as shown in online supplemental figure S1),[14] *Plasmodium falciparum* infection (online supplemental figure S2),[15] and child growth failure, including underweight conditions (online supplemental figure S3).[16] An association between these factors and iNTS disease has been observed in multiple sSA countries. For example, iNTS disease was more common in children with HIV in Kenya[17] and adults in Malawi[18] compared with HIV-uninfected persons. Similarly, children suffering from malnutrition in countries like Kenya,[17] Mozambique,[19 20] Ghana[21] and Tanzania[22] showed higher instances of iNTS disease compared with children without malnutrition. An association between iNTS disease and malaria has been widely reported, including in studies from Malawi[23] and Tanzania.[22] We also explored the impact of environmental factors, access to improved water sources (online supplemental figure S4) and sanitation facilities (online supplemental figure S5),[24] on iNTS disease. This set of covariates was similar to that used in modelling the global burden of iNTS disease.[1]

The HIV prevalence represented estimates of HIV prevalence among adults aged 15–45 years in sSA, annually from 2000 through 2017, based on the review of various surveys including AIDS Indicator Survey, Demographic and Health Survey, Multiple Indicator Cluster Survey, Population-based HIV Impact Assessment Survey and sentinel surveillance of women attending antenatal care clinics. The mean estimates were downloaded from the website of the Global Health Data Exchange (GHDx).[25] The *P. falciparum* parasite rate represents the proportion of children aged 2–10 years showing detectable *P. falciparum* parasite in a given year from 2000 through 2020 and is estimated based on national surveys such as the Demographic and Health Survey data and the literature. The mean estimates were downloaded from the website for the Malaria Atlas Project.[26] Childhood growth failure measured the proportion of children with height-for-age, weight-for-height and weight-for-age (expressed as stunting, wasting and underweight, respectively) *z* score that was more than two SDs below the WHO's median growth reference standards for a healthy population. The mean estimates for 2000–2017 were downloaded from the GHDx.[27] Access to improved drinking water sources and access to improved sanitation facilities represent the proportion of the population who have access to alternative levels of water and sanitation (eg, access to piped water or access to improved water) from 2000 to 2017. The mean estimates were downloaded from GHDx.[28] All covariates were available at the resolution of 5 km × 5 km grids and were aggregated to 20 km × 20 km grids by computing the mean.

We matched the year of data collection and the year for geospatial covariates as closely as possible. All covariates were available for each year from 2000 to 2017 and were matched to occurrences observed in the same year. For occurrences observed before 2000, we used the covariates for the year 2000. For occurrences observed after 2017, we used the covariates for 2017 for water, sanitation and childhood growth failure, as these covariates were only available up to 2017. However, we were able to use the covariates for malaria and HIV through 2020, so we matched the years between the occurrences and covariates for those covariates. Additionally, we matched the location of the data collection and the covariates at 20 km × 20 km grids.

## Probability of occurrence

In this analysis, the outcome is the probability of iNTS occurrence estimated on 20 km × 20 km grids, with each grid serving as the unit of analysis. This approach parallels the classic logistic regression, where the log odds of an event's probability (in this case, probability of iNTS occurrence) within a grid is modelled based on a set of potential risk factors (ie, predictors). These predictors, which are geospatial covariates mentioned previously, were also structured on 20 km × 20 km grids. However, a key distinction arises in how these predictors are combined. Unlike logistic regression, which employs a linear combination of predictors, this model does not assume a linear relationship between covariates and outcome variables and can characterise complex interactions by using the boosted regression tree (BRT) model.[29] The BRT model integrates the strengths of regression trees, where the relationship between an outcome and predictors is modelled through recursive binary splits, and boosting, where many models are combined to improve predictive performance. The BRT framework could use background (or pseudo-absence) data as a substitute for true absence data. Background data[30] represented random grid cells and were sampled such that the probability of being selected increased with decreasing travel time to the healthcare facilities based on the travel time to healthcare facilities[31] (online supplemental figure S6). This choice of background sampling strategy may mitigate a potential bias in our data set that may over-represent occurrences in areas near healthcare facilities. In the BRT, each regression tree includes only explanatory covariates that improve predictions when added over the existing regression trees.

We measured the relative importance of a covariate as proposed in the previous study[29] by measuring how often a variable is selected for splitting individual trees, weighted by the squared improvement to the model as a result of each split, and averaged over all trees.

We grouped the covariates into five categories: HIV, malaria, child growth failure, water and sanitation. Several variants (eg, access to improved water source vs access to piped water) were available for all categories except for HIV (online supplemental table S1). We chose one variable from each category that has the highest influence on the prediction of iNTS occurrence based on the relative importance of covariates from the model containing all covariates to improve interpretability considering that covariates in the same category often have high correlation (eg, *Plasmodium falciparum* parasite prevalence and incidence rate has Pearson's *r* of 0.95, see online supplemental table S1 for details).

## Model fitting and validation

Key parameters in the BRT model include learning rate (LR), tree complexity (TC) and the number of trees. LR, also known as the shrinkage parameter, determines the contribution of each tree to the growing model. The TC controls whether interactions are fitted with a model with a TC of one fitting an additive model, and a TC of two fitting a model with up to two-way interactions, and so on.[29] We determined the values of these two parameters (ie, LR of 0.05 and TC of 5) through a grid search in which the number of trees was optimised through 10-fold cross-validation for each set of TC and LR by taking the model performance and computation time into account. For example, we chose TC of 5 over TC of 10 by considering increasing the TC to 10 improved prediction by 2% but the computation time increased by a factor of two. In the 10-fold cross-validation, the dataset was randomly divided into 10 subsets and models were trained using 9 subsets and were tested against the 10th subset. This process was repeated for each of the 10 subsets and the number of trees was optimised by minimising average prediction deviance. The set of hyperparameters with an optimised number of regression trees that produced the lowest deviance was selected as the final model. We identified the final model for each of the 400 different data sets.

The predictive capacity of the final model was evaluated by calculating the receiver operating characteristic (ROC) curve and an area under the ROC curve (AUC).[32] The models were implemented in statistical software R (V.4.1.3). Raster images were processed for modelling using the raster package and the BRT model was implemented using gbm and dismo packages. All figures were prepared using ggplot2 package. All the codes and datasets are available on GitHub.[33]

## Summarising results

Because of the random nature of the occurrence data (eg, the location for the occurrence reported in the hospital was randomly assigned by the decay function), we created 20 occurrence data sets. For each of the occurrence data sets, we created 20 different background data sets. The probability of occurrence (mean, 2.5th percentile and 97.5th percentile) was predicted by summarising these 400 BRT model simulations based on 400 different data sets for every 20 km × 20 km grid cell. The probability of occurrence was aggregated to the administrative units (eg, districts and provinces) as well as the 20 km × 20 km grid cells. The administrative unit areas were defined based on shapefiles from the Global

A

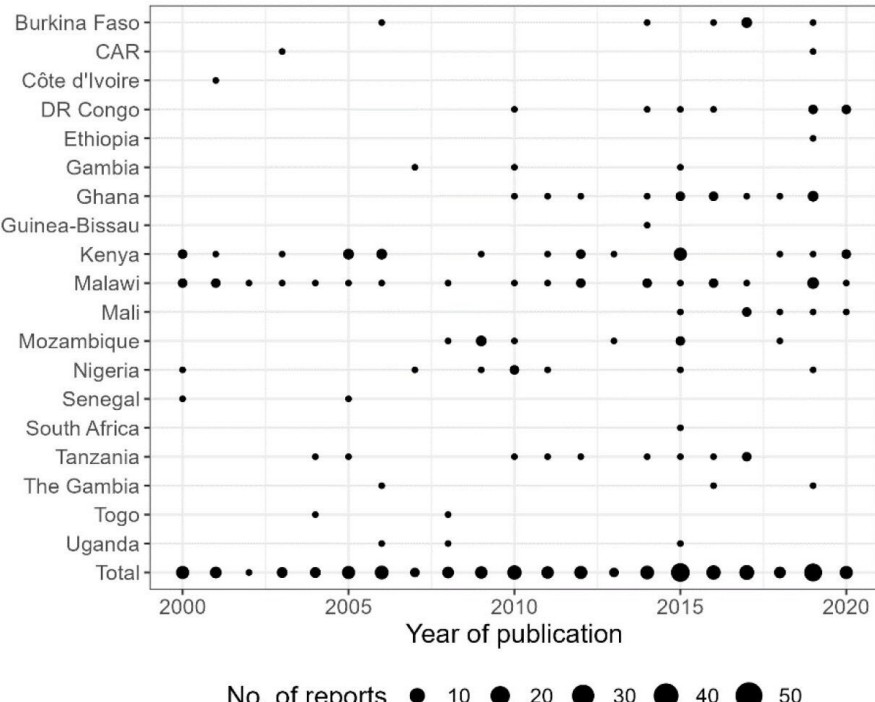

B

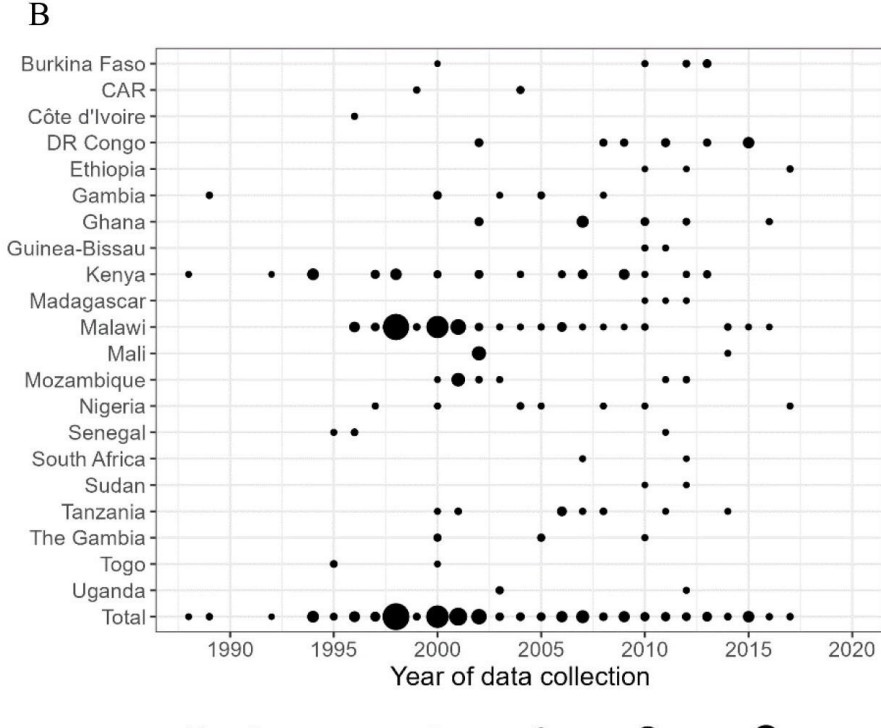

**Figure 2** Data of human invasive non-typhoidal Salmonella infection, sub-Saharan Africa, 2000–2020. (A) represents the number of reports by country and year of publication. (B) represents the number of reported cases by country and year of data collection. CAR, Central African Republic; DR Congo, Democratic Republic of the Congo.

Administrative Areas website.[34] We calculated annual estimates of the probability of occurrence for each year from 2000 to 2017, using the covariates specific to each year. Additionally, we produced forecasts for the years 2018–2020. During this period, the malaria parasite rate was the only covariate that varied annually. The other four covariates were kept constant, using their values as observed in 2017.

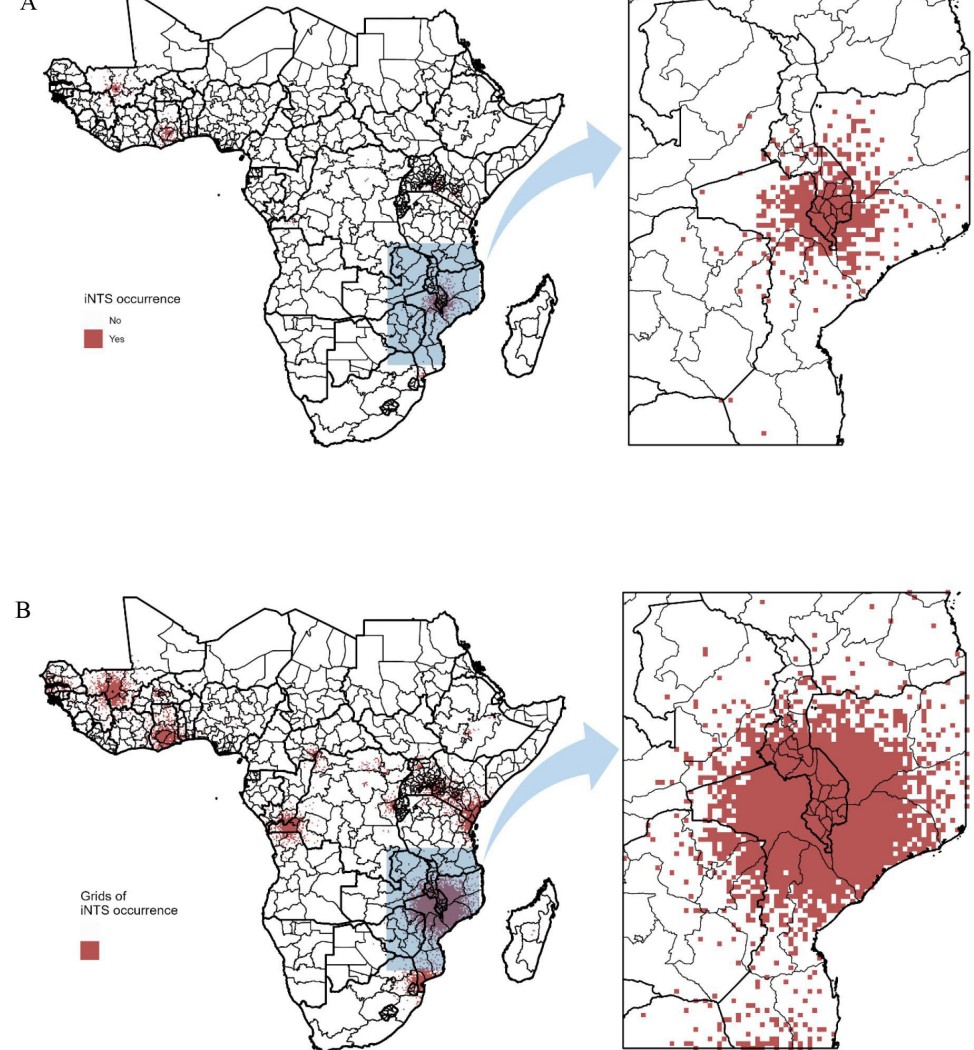

**Figure 3** Invasive non-typhoidal Salmonella (iNTS) occurrence on 20 km × 20 km grids. (A) and (B) represent typical occurrences observed in 2010 and during the entire search period (2000–2020), respectively. Thick and thin lines indicate the borders of the administrative unit level 0 (ie, country) and the administrative unit level 1 (ie, province).

## Patient and public involvement

This study does not involve any human participants.

## RESULTS

We identified 131 unique articles reporting cases of human iNTS disease. Years of publication ranged from 2000 through 2020 (figure 1) while the study years ranged from 1979 through 2020. The number of reports was heterogeneous across year and country with overall 31 016 iNTS disease occurrences reported from 19 countries, and no report of iNTS disease occurrence in the remaining 26 countries of sSA, which does not necessarily mean the absence of occurrences (figure 2A). Of iNTS disease occurrence reports, 25 (19%) were from Kenya, 25 (19%) from Malawi, 13 (10%) from Ghana, 10 (8%) from Tanzania and 58 (44%) from other African countries. Overall, 42 699 iNTS disease cases for which year and location were clearly identified were reported from 2000 through 2020 with variation by year and country (figure 2B). Malawi was the most substantial contributor (n=19 075, 62%) followed by Kenya (n=3539, 11%), Mali (n=2215, 7%) and Mozambique (n=1947, 6%). A majority of studies (n=123, 94%) included blood culture for confirmation of iNTS and the next most common diagnostic methods were culture of cerebrospinal fluid culture (n=14, 11%) followed by culture of pleural (n=2, 2%) or peritoneal (n=1, <1%) fluids. Around 67% (n=88) of the iNTS disease occurrence reports were from tertiary hospitals while about 13% (n=17) came from the primary hospitals and 20% (n=26) of the reports clearly indicated the number of administrative units from which the patients came. These reports led to an average of 5504 iNTS disease occurrences on 20 km × 20 km grids, representing around 3% of 180 490 grid cells comprising sSA. A majority of occurrences focused on the same set of countries (ie, Malawi, Kenya, Mali and Mozambique) that contributed most in terms of the number of cases (figure 3).

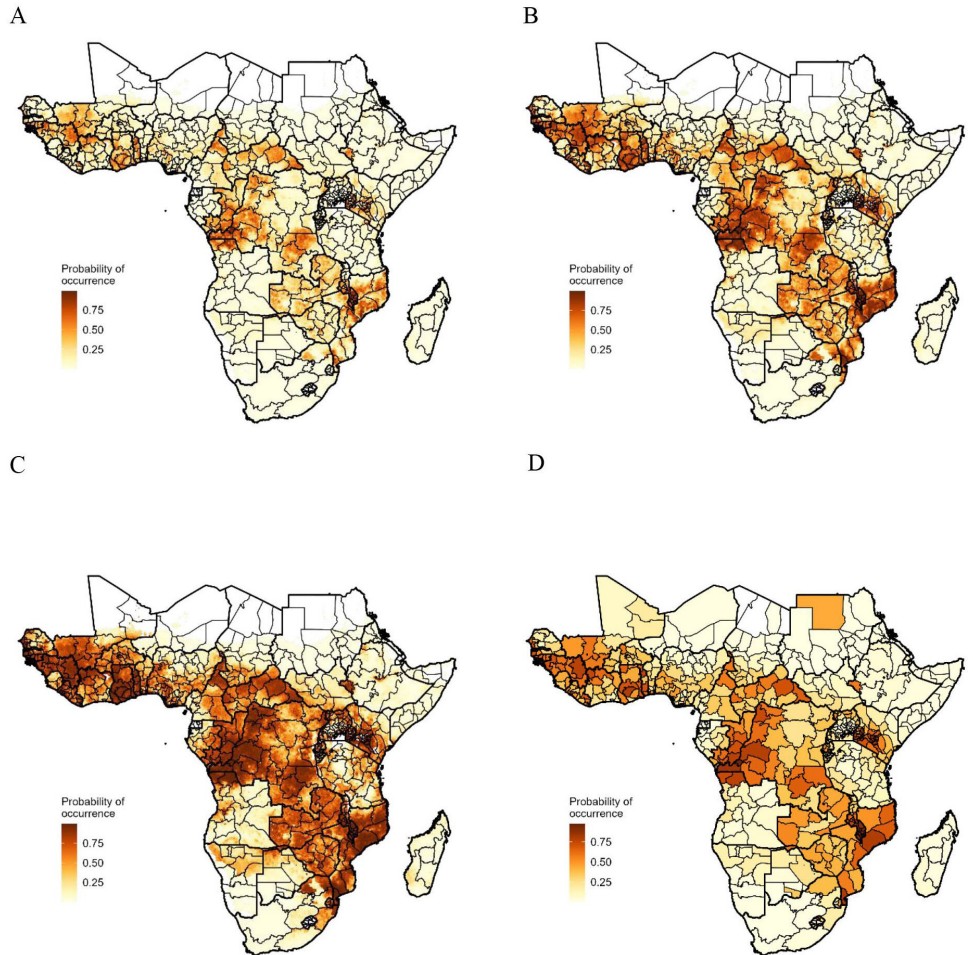

**Figure 4** Estimated probability of invasive non-typhoidal Salmonella disease occurrence, sub-Saharan Africa, 2017. Thick and thin lines indicate the borders of the country and province, respectively. (A), (B) and (C) show mean, 2.5th and 97.5th values on 20 km × 20 km grids. (D) shows the median values aggregated at the administrative unit level 1 (ie, province or districts).

## Probability of iNTS occurrence

Of the 11 geospatial covariates that we explored, we selected five for the final model: *P. falciparum* incidence rate, access to improved water, access to piped sanitation, prevalence of HIV and prevalence of underweight (online supplemental table S1). The Pearson correlation coefficient (*r*) among those five covariates ranged from 0.08 to 0.57. The BRT model predicted well the occurrence of iNTS with the ROC curve indicating an AUC score of around 0.9 (online supplemental figure S7). The estimated probability of iNTS occurrence on 20 km × 20 km grids for 2017 showed significant spatial heterogeneity (figure 4A–C). The probability of occurrence of iNTS aggregated at the subregional level was highest in Middle Africa (0.34, 95% CI: 0.25 to 0.46) followed by Western Africa (0.33, 95% CI: 0.23 to 0.44), Eastern Africa (0.24, 95% CI: 0.17 to 0.33) and Southern Africa (0.08, 95% CI: 0.03 to 0.11) while high-risk grid cells appear to be scattered across the continent. Aggregated at the country level, the probability of occurrence was highest in Malawi and lowest in Niger (online supplemental table S2). The probability of occurrence averaged across administrative

unit level 1 (ie, province) showed similar geospatial variations (figure 4D).

## Contribution of covariates

Geospatial covariates showed varying contributions with underweight having the highest contribution (27.2% (95% CI: 26.2% to 28.1%)) followed by HIV (23.8% (95% CI: 23.1% to 24.5%)), malaria (20.1% (95% CI: 18.4% to 21.6%)), sanitation (18.7% (95% CI: 17.6% to 20.3%)) and drinking water (10.2% (95% CI: 9.6% to 11.0%)) (figure 5A). The relationship between the probability of iNTS disease occurrence and the covariate proved to be complex (figure 5B). The overall relationship between the probability of occurrence and the risk factors such as *P. falciparum* incidence rate, HIV prevalence and underweight prevalence appeared to have a positive relationship.

## Temporal heterogeneity of the probability of occurrence

The spatial distribution of the probability of occurrence varied by year. For 2000, the probability of occurrence was predicted to be highest around Malawi with the country-aggregated value being 0.88 (95% CI: 0.77 to

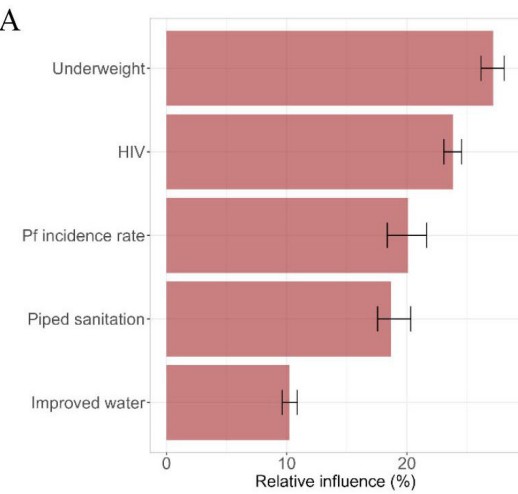

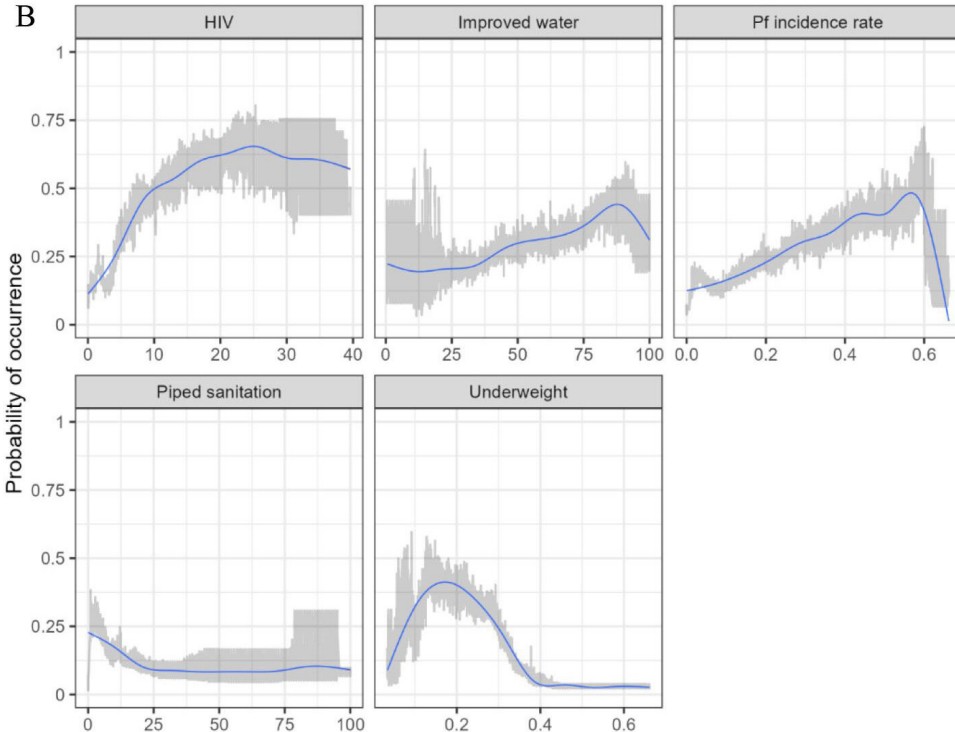

**Figure 5** Impact of covariates on the probability of occurrence. Results are based on 400 simulation runs. (A) represents the relative importance of covariates on the probability of occurrence. Bar plots and error bars indicate the mean with 95% CIs. (B) represents the probability of occurrence in response to variables. The units for sanitation, water and HIV prevalence are expressed as percentages, ranging from 0% to 100%. In contrast, *Plasmodium falciparum* (Pf) incidence and underweight prevalence are measured as proportions, with values ranging from 0 to 1. Smoothed line (blue) and 95% CI (red) bands were based on LOESS (LOcally Estimated Scatterplot Smoothing).

0.94). The next was Mozambique (0.77; 95% CI: 0.62 to 0.87) followed by Uganda (0.65; 95% CI: 0.43 to 0.82) and Zimbabwe (0.62; 95% CI: 0.35 to 0.83) (figure 6A,D). In 2010, Mozambique (0.67; 95% CI: 0.47 to 0.83) had the highest probability of occurrence. Countries in Western Africa such as Ghana (0.64; 95% CI: 0.43 to 0.81), Guinea (0.56; 95% CI: 0.34 to 0.75), Guinea-Bissau (0.52; 95% CI: 0.33 to 0.71) and Liberia (0.49; 95% CI: 0.29 to 0.71) had a high probability of occurrence, with the largest increase between 2000 and 2010 observed in Liberia (0.07 vs 0.49) (figure 6A, B and D). From 2000 through 2020,

the probability of occurrence aggregated at the country level increased most in the Republic of the Congo (0.05 vs 0.59) while decreased most in Uganda (0.65 vs 0.23) (figure 6C,D).

## DISCUSSION

We examined the geospatial and temporal distribution of the probability of occurrence of iNTS for sSA on 20 km × 20 km grids using the data collated from an extensive literature review and high-resolution geospatial covariates

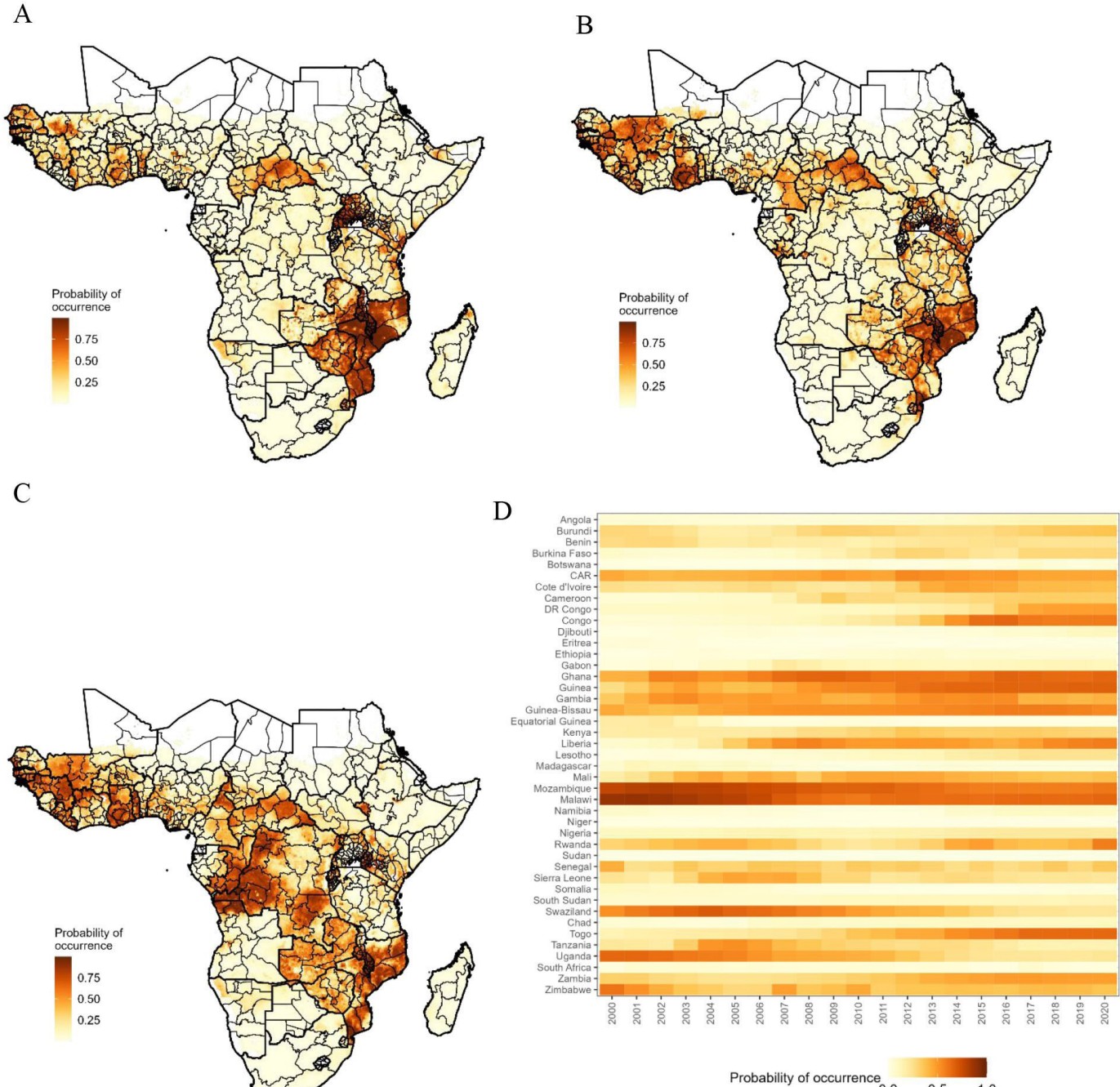

**Figure 6** Probability of occurrence at the 20 km × 20 km grid level and averaged across the country level. (A), (B) and (C) represent the mean probability of occurrence for 2000, 2010 and 2020, respectively. (D) represents the values aggregated at the country level for 2000–2020. CAR, Central African Republic; DR Congo, Democratic Republic of the Congo; Congo, Republic of the Congo.

representing potential risk factors for iNTS disease. The estimated probability of iNTS disease occurrence showed substantial geospatial heterogeneity on 20 km × 20 km grid cells, subnational and national levels, and also temporal variation when measured by year. In 2017, the year for which most recent estimates for the modelled covariates were available, the estimated probability of occurrence was higher in Middle Africa and Western Africa and lower in Eastern Africa and Southern Africa.

Our study is the first to characterise the probability of its iNTS occurrence at the sub-national levels. While predicted incidence rates at the national level are available,[1] these predictions were based on a small number of surveillance studies. Having a means to identify subnational high-risk areas using occurrence data as well as surveillance data is an important addition to the existing knowledge of iNTS epidemiology. The findings may be useful for designing effective and efficient intervention

programmes. Candidate vaccines for iNTS are under active development[35] and deployment strategies are critical for successful intervention and high return on investment. The current study may help identify areas of potential high burden and target for vaccination. Geospatial variation may vary over time (online supplemental figure S8 and table S3) and it will be necessary to keep updating the estimates using more recent information. In a similar vein, our study may help identify areas lacking data for improved surveillance. While surveillance studies provide the most reliable disease occurrence and incidence data, they are resource-intensive, and it is important to conduct surveillance in the areas that could provide the most valuable information such as the Middle Africa (eg, Democratic Republic of the Congo and the Republic of the Congo) in which the occurrence data are scarce while the probability of occurrence was high for years since round 2016 (figure 6D, online supplemental figure S8 and table S4).

Our modelling analyses indicate that geospatial estimates of the host risk factors such as infection with HIV, malaria or underweight are useful for predicting the geospatial distribution of iNTS occurrence while water and sanitation status may also have some smaller predictive capacity. In this regard, developing a risk index of iNTS based on a similar set of variables to what we have used has been attempted for nine sub-Saharan countries.[36]

Our study had several limitations. First, the data points were few, and especially we lacked data on the absence of iNTS disease. While occurrence data points greatly expanded those available from population-based incidence studies, cumulative occurrences from 2000 through 2020 only represented around 3% of the $20\,km \times 20\,km$ grids of sSA. A limited number of countries, primarily Malawi, contributed the majority of the data, with some countries reporting none or few cases, potentially contributing to bias. Nonetheless, our model demonstrates high accuracy, indicated by an ROC curve area of 0.91 (online supplemental figure S7). The predictions (figure 4) show areas with high iNTS disease probability, such as the northern Central African Republic, western Zambia and Cameroon, despite having little or no reported cases. Second, the reported location of iNTS disease occurrence was restricted to hospital-level or high-level administrative units and does not accurately reflect the location of NTS transmission. Hence, the estimated probability of occurrence might not reflect the true probability of iNTS pathogen transmission. Our approach was to assign a catchment area to the hospital and assume that occurrence could happen on any of the grid cells according to a decay function representing a decreasing probability of visits with increasing distance from the hospital. This resulted in a data set that reflected more frequent occurrences in areas close to the hospital than in areas far away. To mitigate this potential bias, we selected background points biased towards shorter distances to healthcare facilities. While occurrences were randomly located based on the exponential decay function, the geospatial and temporal patterns appeared to be robust against different parameterisations for the decay function (online supplemental figure S9). In addition, iNTS disease represented only a fraction of transmission of the pathogen[37] and thus it is likely that the estimates based on the disease will underestimate the true transmission of the pathogen. Third, the frequency of occurrence is likely to be influenced by the frequency of publication in that occurrence is only possible when they are reported in the peer-reviewed journal. Therefore, areas in which iNTS was not studied or reported for any reason were under-represented. Therefore, studies such as population-based surveillance will need to verify the inferred probability of occurrence. Fourth, although the BRT is a machine learning method known for its effective predictive capabilities, interpreting the association between explanatory covariates and the outcome, probability of occurrence of iNTS disease, remains challenging, as illustrated in figure 5B. However, the findings on the variables are generally consistent with our knowledge on the risk factors for iNTS disease and we could still combine the knowledge on the probability of occurrence and the potential risk factors to prioritise our resources for more effective control of disease.

## Conclusion

Our analysis suggests that the occurrence of iNTS disease varies greatly across sSA, with certain regions being disproportionately affected. This geospatial heterogeneity is crucial to understand to effectively combat the iNTS disease. By identifying areas with a high probability of occurrence, targeted surveillance can be conducted to better understand the incidence and the drivers of the disease in those regions. This information can then be used to develop intervention programmes to prevent the spread of iNTS. Ultimately, a deeper understanding of the geographical distribution of iNTS can help to reduce its impact and improve public health outcomes in sSA.

**Author affiliations**
[1]International Vaccine Institute, Gwanak-gu, Seoul, Republic of Korea
[2]Clinical Sciences, Institute of Tropical Medicine, Antwerp, Belgium
[3]Department of Microbiology, Immunology and Transplantation, KU Leuven, Leuven, Belgium
[4]Department of Medical Biotechnologies, University of Siena, Siena, Italy
[5]Department of Medicine and Surgery, LUM University "Giuseppe Degennaro", Bari, Italy
[6]New Zealand Ministry of Health, Wellington, New Zealand
[7]Department of Clinical Sciences, Institute of Tropical Medicine, Antwerp, Belgium
[8]Centre for International Health, University of Otago, Dunedin, New Zealand
[9]Epidemiology Unit, International Vaccine Institute, Seoul, Republic of Korea
[10]Department of Medicine, University of Cambridge, Cambridge, UK

**Correction notice** This article has been corrected since it was first published. Author names 'John A Crump' and 'Christian S Marchello' have been updated.

**Acknowledgements** We thank the Vacc-iNTS team for technical support and continuous encouragement. Vacc-iNTS consortium collaborators include: Francis Agyapong (Kwame Nkrumah University of Science and Technology Kumasi), Francesco Berlanda Scorza (GSK Vaccines Institute for Global Health), Gianluca Breghi (Fondazione Achille Sclavo), Rocío Canals (GSK Vaccines Institute for Global

Health), Melita A Gordon (University of Liverpool), Brama Hanumunthadu (University of Oxford), Samuel Kariuki (Kenya Medical Research Institute), Stefano Malvolti (MM Global Health Consulting), Carsten Mantel (MM Global Health Consulting), Donata Medaglini (Università di Siena and Sclavo Vaccines Association), Esther Muthumbi (KEMRI-Wellcome Trust Research Programme), Mercy Ngetich (Kenya Medical Research Institute), Tonney S Nyirenda (University of Malawi), Ellis Owusu-Dabo (Kwame Nkrumah University of Science and Technology Kumasi), Maheshi Ramasamy (University of Oxford), J Anthony Scott (KEMRI-Wellcome Trust Research Programme), Bassiahi Abdramane Soura (University of Ouagadougou), Tiziana Spadafina (Sclavo Vaccines Association).

**Collaborators** Vacc-iNTS consortium.

**Contributors** Responsible for the overall content as guarantor: J-HK. Conceptualisation: J-HK, FM, Vacc-iNTS consortium. Data curation: BT, FF, EP, CM. Formal analysis: J-HK. Funding acquisition: FM, Vacc-iNTS consortium collaborators. Investigation: J-HK, BT, JJ, JC. Methodology: J-HK, BT. Resources: J-HK, FM, Vacc-iNTS consortium. Software: J-HK. Supervision: J-HK, FM, Vacc-iNTS consortium. Validation: J-HK, BT, JJ, JC. Visualisation: J-HK. Writing—original draft: J-HK, BT, JJ, JC. Writing—review and editing: J-HK, BT, FF, EP, JJ, JC, FM, Vacc-iNTS consortium.

**Funding** This project has received funding from the EU Horizon 2020 research and innovation programme under the project Vacc-iNTS (grant agreement number 815439).

**Map disclaimer** The inclusion of any map (including the depiction of any boundaries therein), or of any geographical or locational reference, does not imply the expression of any opinion whatsoever on the part of BMJ concerning the legal status of any country, territory, jurisdiction or area or of its authorities. Any such expression remains solely that of the relevant source and is not endorsed by BMJ. Maps are provided without any warranty of any kind, either express or implied.

**Competing interests** None declared.

**Patient and public involvement** Patients and/or the public were not involved in the design, or conduct, or reporting, or dissemination plans of this research.

**Patient consent for publication** Not applicable.

**Provenance and peer review** Not commissioned; externally peer reviewed.

**Data availability statement** Data are available upon reasonable request. Data and codes are available on GitHub.

**ORCID iDs**
Jong-Hoon Kim http://orcid.org/0000-0002-9717-4044
Bieke Tack http://orcid.org/0000-0002-1129-0440
John A Crump http://orcid.org/0000-0002-4529-102X
Florian Marks http://orcid.org/0000-0002-6043-7170

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
