## [Reviewer comments · BMJ Open]

ARTICLE DETAILS

TITLE (PROVISIONAL)	Examining geospatial and temporal distribution of invasive non-typhoidal Salmonella disease occurrence in sub-Saharan Africa: A systematic review and modeling study
AUTHORS	Kim, Jong-Hoon; Tack, Bieke; Fiorino, Fabio; Pettini, Elena; Marchello, Christian; Jacobs, Jan; Crump, John; Marks, Florian; Vacc-iNTS Consortium, Collaborators

VERSION 1 – REVIEW

REVIEWER	Fauzi, Ilham Saiful Politeknik Negeri Malang, Accounting
REVIEW RETURNED	23-Oct-2023

GENERAL COMMENTS	In this paper, the authors analyze the geospatial variation related to occurrence of iNTS in sub-Saharan Africa, examining them at a subnational level with a 20 km x 20 km resolution. This study is intriguing and could serve as a valuable resource for determining priority areas for focused iNTS surveillance and control program. However, I would like to suggest a few enhancements to the paper. 1. Regarding the use of year periods, all mapping starts from 2000, even though the results section mentions that the data for iNTS events spans from 1979 to 2020. From what I understand, the period 2000-2020 aligns with the year of publication of the reference source. Does Figure S8 in the mapping incorporate data from 2000-2020, and what about iNTS occurrences data prior to 2000?2. The mapping results in Figure 3 illustrate an aggregate for the 2000-2020 period. Would it be more informative to display mappings annually to reveal patterns of changing risk areas? This would help us understand if certain areas consistently exhibit high risk or if risk areas tend to shift over time.3. Based on the results presented by the author, is it possible to predict the incidence of iNTS for the years 2021-2025? Such predictions in recent years would provide a clearer picture of areas with a high likelihood of outbreaks, aiding in the optimization of surveillance and control programs.4. The author utilizes several covariates, as mentioned in the geospatial covariates subsection, which are matched with iNTS occurrences data. It would enhance the paper if the author briefly explained the rationale behind selecting these covariates and how they may be associated with the incidence of iNTS.
--

	5. Other comments: a. The author should ensure that references are properly cited within the manuscript, and vice versa. b. All figures and tables should be mentioned in the manuscript. However, Figure S6 is not referenced, making its purpose unclear. c. Figures 6 and Figure S8 present nearly identical information, with the only difference being the years. To enhance the effectiveness of these figures, it is advisable to maintain consistency in the choice of years, whether they are annual, biennial, five-year intervals, or another appropriate option. d. In Figure 5B, it would be helpful to label the x-axis to clarify the meaning of the numbers displayed. e. Figures S2, S4, and S5 depict the northern area without color. Is this intentional, and does it accurately represent the data? f. Regarding Table S1, please review the yellow coloring, and consider explaining why certain highly correlated coefficients were not selected as covariates, such as stunting. g. The writer should ensure that the manuscript is free of typographical and grammatical errors.
--	---

REVIEWER	Varga, Csaba University of Illinois at Urbana-Champaign, Pathobiology
REVIEW RETURNED	05-Nov-2023

GENERAL COMMENTS	I have enjoyed reading your article on invasive non-typhoidal Salmonella disease occurrence in sub-Saharan Africa. Please find my comments below. 1.) It is not clear to me what is the outcome variable in your analysis. You describe the outcome as the number of reported cases in each 20 km by 20 km grid cell. However, it is required to account for the background population in each grid cell, knowing that the probability of case reports from a large background population is higher than from areas with low populations. 2) Could you clarify what type of model did you use? I assume that your model should be a Poisson or Negative binomial model considering that your outcome is a count if you only report the number of reported cases in each grid cell or incidence rate if you account for the background population. 3) Would you describe how the non-response areas were included in your model? If you included those areas as "0" is not accurate. You should use a spatial interpolation method and estimate the incidence rate or count of those areas based on the disease rate of nearby areas. 4) In my mind, the geospatial heterogeneity could be due to the reporting bias. Almost 50% of cases were reported from one region (Malawi, n=19,075). I suggest running country-specific models. Presenting your finding as relevant to sub-Saharan Africa, it might be an overstatement as several regions were not reporting any case.
--

VERSION 1 – AUTHOR RESPONSE

Reviewer: 1

Dr. Ilham Saiful Fauzi, Politeknik Negeri Malang, Bandung Institute of Technology

Comments to the Author:

In this paper, the authors analyze the geospatial variation related to occurrence of iNTS in sub-Saharan Africa, examining them at a subnational level with a 20 km x 20 km resolution. This study is intriguing and could serve as a valuable resource for determining priority areas for focused iNTS surveillance and control program. However, I would like to suggest a few enhancements to the paper.

1. Regarding the use of year periods, all mapping starts from 2000, even though the results section mentions that the data for iNTS events spans from 1979 to 2020. From what I understand, the period 2000-2020 aligns with the year of publication of the reference source. Does Figure S8 in the mapping incorporate data from 2000-2020, and what about iNTS occurrences data prior to 2000?

The iNTS disease occurrences prior to 2000 were included. Following sentences were adjusted to clarify the issue.

Lines 164-169:

For occurrences observed before 2000, we used the covariates for the year 2000. For occurrences observed after 2017, we used the covariates for 2017 for water, sanitation, and childhood growth failure, as these covariates were only available up to 2017. However, we were able to use the covariates for Malaria and HIV through 2020, so we matched the years between the occurrences and covariates for those covariates. Additionally, we matched the location of the data collection and the covariates at 20-km by 20-km grids.

Lines 234-235:

We identified 131 unique articles reporting cases of human iNTS disease. Years of publication ranged from 2000 through 2020 (Figure 1) while the study years ranged from 1979 through 2020.

2. The mapping results in Figure 3 illustrate an aggregate for the 2000-2020 period. Would it be more informative to display mappings annually to reveal patterns of changing risk areas? This would help us understand if certain areas consistently exhibit high risk or if risk areas tend to shift over time.

Thank you for your suggestion. Figure 3 displays the cumulative grids of potential iNTS occurrences during the study period. Figure 4 shows the estimated probability of iNTS occurrence for 2017, the latest year with available model covariates. Yearly estimates for 2000-2020 are presented in Figure S8. Notably, the estimates for 2018-2020 in Figure S8 were calculated under the assumption that certain covariate values remained constant from 2017 to 2020.

3. Based on the results presented by the author, is it possible to predict the incidence of iNTS for the years 2021-2025? Such predictions in recent years would provide a clearer picture of areas with a high likelihood of outbreaks, aiding in the optimization of surveillance and control programs.

Unfortunately, our model cannot generate incidence data for the years 2021-2025. It is designed to calculate the probability of occurrence of iNTS based on various potential risk factors. To estimate the

probability of occurrence for 2021-2025, we would need covariate data for these years, which we currently do not have.

4. The author utilizes several covariates, as mentioned in the geospatial covariates subsection, which are matched with iNTS occurrences data. It would enhance the paper if the author briefly explained the rationale behind selecting these covariates and how they may be associated with the incidence of iNTS.

We expanded the section that describes the potential risk factor section to make it richer and easier to follow. Now the section (lines 130-143) reads as follows:

Our study investigated factors that may affect the risk of iNTS disease across sub-Saharan Africa, focusing on a variety of host [13] and environmental factors [7]. We examined these factors at a detailed resolution of at least 20 km x 20 km across the continent. For host factors, we looked into the prevalence of HIV (as shown in Figure S1) [14], Plasmodium falciparum infection (Figure S2) [15], and child growth failure, including underweight conditions (Figure S3) [16]. An association between these factors and iNTS disease has been observed in multiple sub-Saharan African countries. For example, iNTS disease was more common in children with HIV in Kenya [17] and adults in Malawi [18] compared with HIV-uninfected persons. Similarly, children suffering from malnutrition in countries like Kenya [17], Mozambique [19,20], Ghana [21], and Tanzania [22] showed higher instances of iNTS disease compared to children without malnutrition. An association between iNTS disease and malaria has been widely reported, including in studies from Malawi [23] and Tanzania [22]. We also explored the impact of environmental factors, access to improved water sources (Figure S4) and sanitation facilities (Figure S5) [24], on iNTS disease. This set of covariates was similar to that used in modeling the global burden of iNTS disease [1].

5. Other comments:

a. The author should ensure that references are properly cited within the manuscript, and vice versa.

We reviewed the references and believe that the references are cited correctly.

b. All figures and tables should be mentioned in the manuscript. However, Figure S6 is not referenced, making its purpose unclear.

We now reference Figure S6 in the section where travel time to healthcare facilities is mentioned.

c. Figures 6 and Figure S8 present nearly identical information, with the only difference being the years. To enhance the effectiveness of these figures, it is advisable to maintain consistency in the choice of years, whether they are annual, biennial, five-year intervals, or another appropriate option.

Figure S8 displays annual data from 2000 to 2020, while Figure 6 highlights key data points from three specific years: 2000, 2010, and 2020, serving as a summary. The rationale for selecting these

particular years is detailed in the section dedicated to summarizing the results. We have enhanced this section for greater clarity (lines 227-230 in the revised version).

We calculated annual estimates of the probability of occurrence for each year from 2000 to 2017, utilizing the covariates specific to each year. Additionally, we produced forecasts for the years 2018 to 2020. During this period, the malaria parasite rate was the only covariate that varied annually. The other four covariates were kept constant, using their values as observed in 2017.

d. In Figure 5B, it would be helpful to label the x-axis to clarify the meaning of the numbers displayed.

The section on geospatial covariates provides detailed information about these variables. However, for brevity in the figure legends, we have included only a brief description of the units for each variable (see below), as their names are already depicted in the figures.

Figure 5. Impact of covariates on the probability of occurrence. Results are based on 400 simulation runs. (A) represents relative importance of covariates on the probability of occurrence. Bar plots and error bars indicate mean with 95% confidence intervals. (B) represents probability of occurrence in response to variables. The units for sanitation, water, and HIV prevalence are expressed as percentages, ranging from 0 to 100%. In contrast, Plasmodium falciparum (Pf) incidence and underweight prevalence are measured as proportions, with values ranging from 0 to 1. Smoothed line (blue) and 95% confidence interval (red) bands were based on LOESS (LOcally Estimated Scatterplot Smoothing).

e. Figures S2, S4, and S5 depict the northern area without color. Is this intentional, and does it accurately represent the data?

The areas depicted are not errors; they represent desert regions where we lack data for both the covariates and the occurrence of iNTS. This information has been added to the legends of the figures in the updated version for clarity.

For example, legends for the Figure S2 now reads:

Figure S2. Incidence rate of Plasmodium Falciparum [2], 2017. The values indicate the number of newly diagnosed Plasmodium falciparum cases per person in 2017. Areas for which data are not available are not colored.

f. Regarding Table S1, please review the yellow coloring, and consider explaining why certain highly correlated coefficients were not selected as covariates, such as stunting.

The selection of covariates was based on the variable's relative importance, as detailed in Lines 193-199 of the revised version. Table S1 illustrates the correlation among these covariates, demonstrating our approach of choosing a single variable from each group of similar correlated variables (e.g., selecting underweight but not both stunting and underweight).

g. The writer should ensure that the manuscript is free of typographical and grammatical errors.

The manuscript was checked for typographical and grammatical errors.

*Please see attached report from this reviewer

Reviewer: 2

Dr. Csaba Varga, University of Illinois at Urbana-Champaign

Comments to the Author:

I have enjoyed reading your article on invasive non-typhoidal Salmonella disease occurrence in sub-Saharan Africa. Please find my comments below.

1.) It is not clear to me what is the outcome variable in your analysis. You describe the outcome as the number of reported cases in each 20 km by 20 km grid cell. However, it is required to account for the background population in each grid cell, knowing that the probability of case reports from a large background population is higher than from areas with low populations.

The outcome variable is defined as the probability of occurrence in each grid cell, not on an individual basis. This probability of occurrence within a grid cell may rise due to factors such as population size (as the reviewer indicated) or other host or environmental factors.

We expanded the section that describes the probability of occurrence to make it easier to follow. Now the section reads as follows (lines 172-183 in the revised version):

In this analysis, the outcome is the probability of INTS occurrence estimated on 20-km by 20-km grids, with each grid serving as the unit of analysis. This approach parallels the classic logistic regression, where the log odds of an event's probability (in this case, probability of INTS occurrence) within a grid is modeled based on a set of potential risk factors (i.e., predictors). These predictors, which are geospatial covariates mentioned previously, were also structured on 20-km by 20-km grids. However, a key distinction arises in how these predictors are combined. Unlike logistic regression, which employs a linear combination of predictors, this model does not assume a linear relationship between covariates and outcome variables and can characterize complex interactions by utilizing the boosted regression tree boosted regression tree (BRT) model [29]. The BRT model integrates strengths of regression trees, where relationship between an outcome and predictors is modeled through recursive binary splits, and boosting, where many models are combined to improve predictive performance.

29. Elith J, Leathwick JR, Hastie T. A working guide to boosted regression trees. *Journal of Animal Ecology*. 2008;77: 802–813. doi:10.1111/j.1365-2656.2008.01390.x

2) Could you clarify what type of model did you use? I assume that your model should be a Poisson or Negative binomial model considering that your outcome is a count if you only report the number of reported cases in each grid cell or incidence rate if you account for the background population.

We employed the boosted regression tree model, as detailed in lines 172-189 of the revised version. Our model is designed to classify binary outcomes, differentiating between presence and absence, akin to the logistic regression model. As previously mentioned in response to an earlier query, our model is not adjusted for background population size.

3) Would you describe how the non-response areas were included in your model? If you included those areas as "0" is not accurate. You should use a spatial interpolation method and estimate the incidence rate or count of those areas based on the disease rate of nearby areas.

In line with our responses to earlier questions, the boosted regression tree model that we implemented was designed to analyze data regarding the presence or absence of iNTS disease. For the absence data, we utilized what is known as pseudo-absence data. These data points were randomly chosen but were representative of the travel time to healthcare facilities, a method employed to balance potential biases in the presence data. The methodology for this process is elaborated in lines 183-187 of the revised version. Additionally, we have included an extra reference (below) to substantiate our use of pseudo-absence data.

Ref: Barbet-Massin M, Jiguet F, Albert CH, Thuiller W. Selecting pseudo-absences for species distribution models: how, where and how many? *Methods in Ecology and Evolution*. 2012;3(2):327–38.

4) In my mind, the geospatial heterogeneity could be due to the reporting bias. Almost 50% of cases were reported from one region (Malawi, n=19,075). I suggest running country-specific models. Presenting your finding as relevant to sub-Saharan Africa, it might be an overstatement as several regions were not reporting any case.

Thank you for highlighting a crucial aspect. Our study sought to maximize the use of available data, including occurrence data as well as incidence rates, to understand the geographical distribution of iNTS disease in sub-Saharan Africa. We used background or pseudo-absence data to counteract reporting bias, for example where iNTS disease episodes near healthcare facilities might be over-reported. While Malawi contributed most data and some countries reported no or few cases, leading to potential bias, our model nonetheless shows good accuracy with an ROC curve area of 0.91 (Figure S7). The predictions (Figure 4) indicate high iNTS probability in areas like the northern Central African Republic or Western Zambia or Cameroon, despite no or few reported cases. Country-specific models do not align with our goal of mapping iNTS distribution across the entire region.

We have updated the Discussion section to reflect the above discussions (lines 319-324 of the revised version).

...

A limited number of countries, primarily Malawi, contributed the majority of data, with some countries reporting none or few cases, potentially contributing to bias. Nonetheless, our model demonstrates high accuracy, indicated by an ROC curve area of 0.91 (Figure S7). The predictions (Figure 4) show areas with high iNTS disease probability, such as northern Central African Republic, western Zambia, and Cameroon, despite having little or no reported cases. ...

VERSION 2 – REVIEW

REVIEWER	Fauzi, Ilham Saiful Politeknik Negeri Malang, Accounting
REVIEW RETURNED	07-Feb-2024

GENERAL COMMENTS	The authors have revised the manuscript according to my suggestions for improvement. All questions were also answered satisfactorily. Overall I recommend this article for publication.
---

REVIEWER	Varga, Csaba University of Illinois at Urbana-Champaign, Pathobiology
REVIEW RETURNED	03-Feb-2024

GENERAL COMMENTS	Thank you for addressing all of my suggestions!
---

VERSION 2 – AUTHOR RESPONSE